# Prevalence and factors associated with illicit substance use among persons with Schizophrenia at a Tertiary Referral Hospital in Zambia

Kimberley R. Kurehwatira[1,2‡], Emmanuel L. Luwaya[1,2‡], Joreen P. Povia[3], Chileleko Siakabanze[2], Salma M. Baines[1,2], Emmanuel Yumba[1,2], Prince Mulambo[1,2], David N. Masta[1,2], Nestorine N. Ngongo[1,2], Natasha Chishala[1,2], Emmanuel O. Riwo[1,2], Katongo H. Mutengo[2], Hanzooma Hatwiko[1,2], Martin Chakulya[1,2], Lukundo Siame[1,2], Bislom C. Mweene[1,2], Sepiso K. Masenga[1,2*]

1 Department of Pathology, Mulungushi University, School of Medicine and Health Sciences, Livingstone, Zambia, 2 Department of Pathology, Livingstone Center for Prevention and Translational Science, Livingstone, Zambia, 3 Department of Health Economics, Livingstone Center for Prevention and Translational Science, Livingstone, Zambia

‡ Share co-first authorship.
* sepisomasenga@gmail.com, sepisomasenga@lcpts.org

## Abstract

Schizophrenia frequently involves comorbid substance use, exacerbating symptoms and reducing treatment efficacy yet no prior studies have examined this comorbidity locally. This study determined the prevalence and factors associated with illicit substance use among adults with schizophrenia at the Livingstone University Teaching Hospital (LUTH). A hospital-based single center cross-sectional study was conducted at LUTH from January to June 2023. A sample of 303 adults with schizophrenia were recruited via systematic random sampling. Data were collected from existing medical records which included documentation from structured interviews with validated tools (Alcohol, Smoking, and Substance Involvement Screening Test [ASSIST], Positive and Negative Syndrome Scale [PANSS]) conducted as part of routine clinical care. Sociodemographic, clinical, and substance use variables were analyzed using chi-square tests and multivariate logistic regression to identify factors associated with illicit substance use. The prevalence of illicit substance use was 31.1% (94/303). Alcohol use (AOR = 6.08, 95% CI: 3.14–11.78, $p < 0.0001$) and tobacco smoking (AOR = 4.80, 95% CI: 2.44–9.46, $p < 0.0001$) were strongly associated with illicit substance use. Factors associated with lower odds of illicit substance use included female sex (AOR = 0.27, 95% CI: 0.12–0.60, $p = 0.001$), marriage (AOR = 0.39, 95% CI: 0.19–0.79, $p = 0.008$), having both parents deceased (AOR = 0.29, 95% CI: 0.11–0.75, $p = 0.011$), and higher education (AOR = 0.44, 95% CI: 0.24–0.81, $p = 0.009$). Increased hospitalizations were associated with illicit substance use (AOR = 1.29, 95% CI: 1.01–1.65,

**Data availability statement:** The raw data underlying the results presented in the study have been uploaded as supporting information.

**Funding:** The authors received no specific funding for this work.

**Competing interests:** The authors have declared that no competing interests exist.

p = 0.038). Integrated screening, gender-specific interventions, and socioeconomic support related to illicit substance use are urgently needed in Zambia's resource-constrained setting.

---

## Introduction

Schizophrenia, a debilitating psychiatric disorder affecting approximately 24 million people globally (WHO, 2022) [1], is characterized by profound disruptions in cognition, emotion, and behaviour [2]. In sub-Saharan Africa (SSA), schizophrenia is compounded by systemic challenges such as poverty, stigma, and fragmented healthcare systems. Comorbid substance use, prevalent in 40–60% of schizophrenia cases globally [3], exacerbates disease burden, impairing treatment adherence and increasing relapse rates. In SSA, this comorbidity is particularly severe, with studies reporting substance use rates as high as 58–65% among persons with schizophrenia [3]. Zambia, a low-income country with a population of 22 million, faces acute mental health challenges: mental disorders account for 13% of the national disease burden, yet the country has only ten psychiatrists as reported by WHO in 2023 [4], a slight increase from five psychiatrists in 2017, and limited mental health funding [5]. The University Teaching Hospital (LUTH), Zambia's largest referral center, struggles with high patient loads and resource constraints, making it a critical site for studying schizophrenia and substance use.

Substance use patterns are shaped by cultural, economic, and environmental factors [6–11]. The widespread use of *kachasu* (a cheap, a traditionally fermented, highly intoxicating, distilled alcoholic spirit), cannabis, and emerging synthetic drugs is normalized in many communities. Economic hardship affecting 52.4% of Zambians who live below the poverty line [12] and unemployment (13.1% nationally) drive maladaptive coping behaviours, including substance use. Traditional beliefs attributing mental illness to spiritual causes further delay biomedical care, pushing patients toward self-medication with substances [10]. Despite these challenges, no studies have systematically examined the prevalence or correlates of illicit substance use among persons with schizophrenia in Zambia. This gap impedes the development of culturally tailored interventions, perpetuating cycles of poor outcomes.

Zambia's mental health system is strained: mental health receives <1% of the health budget, and community stigma forces many patients into isolation [12]. Illicit substance use among persons with schizophrenia at LUTH remains unstudied, yet it likely fuels high relapse rates and recurrent hospitalizations. The study determined the prevalence and factors associated with illicit substance use among adults with schizophrenia at the University Teaching Hospital (LUTH) in Zambia. We hypothesized that male sex, alcohol use, and tobacco smoking would be associated with higher odds of illicit substance use, while factors like higher education and being married would be associated with lower odds.

## Methodology

### Ethics statement

Ethical approval was obtained from the Mulungushi University School of Medicine and Health Sciences Research Ethics Committee (ethics reference number SMHS-MU1-2025-42) on 24 March 2025. Administrative permission was granted by LUTH management. All extracted data were anonymized to protect confidentiality. All data collected and analyzed were de-identified to ensure complete confidentiality. No information leading to identification of patients during and after analysis was abstracted and entered in the data collection form. Secondary data were used in this project. A written/verbal consent was not applicable and was therefore waived by the ethics committee.

### Study design

This was a retrospective cross-sectional study. We used existing medical records from persons with schizophrenia at LUTH in Zambia.

### Study setting

The research was conducted at the Psychiatry Department of LUTH, a tertiary referral hospital located in Livingstone, Zambia. The department systematically documents clinical features, substance use history, and treatment of persons diagnosed with schizophrenia.

### Study population

**Target population.** Adults (≥18 years) with schizophrenia in Livingstone.
**Accessible population.** Persons with Schizophrenia seeking care at LUTH's Psychiatry Department (between January 2021 and December 2024).

### Actual study population

303 patients meeting inclusion criteria, representing 36.1% of screened records (303/840). All subtypes of schizophrenia (ICD-10: F20.0–F20.9) were included.

A post-hoc power analysis was conducted using G*Power 3.1 to evaluate the statistical power of our study design. This analysis revealed 99.8% power at $\alpha = 0.05$ to detect the observed effect size (OR=6.79) for the primary variable of male sex. However, for rare outcomes such as suicidality, where only three cases were observed, the power was substantially lower at just 12% to detect meaningful associations. The power calculation for male sex specifically employed a two-proportion z-test for independent samples with a two-tailed design. This test compared proportions of 0.448 (substance use in males) and 0.107 (substance use in females), with sample sizes of 181 males and 122 females at the conventional alpha level of 0.05. These parameters confirmed robust power for our main sex-based comparison while highlighting limitations for analyzing infrequent outcomes.

### Eligibility and recruitment

We reviewed medical records of persons diagnosed with schizophrenia who received care at LUTH between January 2021 and December 2024. The diagnosis of schizophrenia was made by a consultant psychiatrist according to ICD-10 criteria. Both inpatients and outpatients were included, regardless of their clinical remission status. Records were included if the patient was 18 years or older, had a confirmed psychiatrist-diagnosed schizophrenia (ICD-10 F20), and the medical record contained complete documentation of sociodemographic data and substance use history. A total of 303 patient records were included in the final analysis. Records were included if the patient had a confirmed diagnosis of

schizophrenia and complete documentation of sociodemographic and substance use information. Records missing key variables were excluded.

## Data collection

Data were collected through a structured file review process between 1st April and 28 April 2025. Trained research assistants extracted relevant data from patients' medical files into a REDCap database. No patient interviews were conducted. Information from structured clinical interviews (e.g., ASSIST, PANSS) had been previously conducted by clinic staff and documented in the records; this historical data was extracted for analysis. To ensure data quality, 10% of the records were subjected to dual review for accuracy and consistency, and performed an audit trail using RedCap audit logs to track all data entries and revisions. Additionally, records missing variables were excluded. Other quality measures included the training of personnel involved in data collection using mock records and piloting on records to refine variable definitions.

## Assessment of clinical variables

Clinical variables were ascertained from the psychiatrist's notes and assessment scales documented in the medical records. Hallucinations, delusions, disorganized speech, disorganized behavior, and negative symptoms were assessed and documented by the treating psychiatrist based on clinical evaluation and, where available, PANSS scores. Cognitive impairment was noted based on clinical observation and any documented cognitive assessments. Suicidality and self-harm were defined by documented reports of suicidal ideation, plans, or attempts, or acts of non-suicidal self-injury. Violent behavior was defined as documented episodes of physical aggression towards others or property.

## Study variables

The primary outcome variable was current illicit substance use, defined as documented use of psychoactive substances (cannabis, cocaine, opioids, or others) within the past 12 months.

Independent variables included: Sociodemographic Variables (Age, Sex, Marital status, Educational level, Employment status, Residence, Parental status) and Clinical Variables (hallucinations and type, delusions and type, disorganized speech, disorganized behavior, negative symptoms, cognitive impairment, suicidality and self-harm, violent behavior, family history, tobacco smoking status, awareness of diagnosis, consultation with traditional healers, medication adherence, number of hospitalizations, hematological and biochemical parameters).

## Data analysis

Data were exported from REDCap into Microsoft Excel 2013 for cleaning and coding, then analyzed using StatCrunch (StatCrunch, LLC, Pearson's web-based statistical software). Descriptive statistics summarized categorical variables (frequencies and percentages) and continuous variables (medians and interquartile ranges [IQR]). Normality was assessed using the Shapiro-Wilk test. Associations between categorical variables were evaluated using the Chi-square test, while differences between continuous variables were assessed using the Wilcoxon rank-sum test. Univariable and multivariable logistic regression analyses were performed to identify factors associated with current illicit substance use. Adjusted odds ratios (AORs) with 95% confidence intervals (CIs) were reported. A p-value of $< 0.05$ was considered statistically significant.

Variable selection for multivariable logistic regression required balancing statistical rigor with clinical relevance and practical constraints. Key steps included an initial screening phase using liberal p-values (e.g., $< 0.20 - 0.25$), followed by final variable inclusion based on stricter statistical significance ($p < 0.05$). Crucially, clinically important variables were mandatorily considered for inclusion regardless of their statistical p-value. Potential multicollinearity among variables was assessed using the Variance Inflation Factor (VIF $< 10$), absolute correlation coefficients ($|r| < 0.8$), and condition index ($<$ 30). Sample size sufficiency was ensured, ideally targeting a minimum of 10–15 + events per variable. Various selection

methods (e.g., Forward, Backward, Stepwise, Best Subset) were employed, guided by criteria such as p-values or information criteria like the Akaike Information Criterion (AIC) and the Bayesian Information Criterion (BIC). The final model was derived using backward elimination, and variables significant in bivariate analysis but not included in the final model (e.g., negative symptoms, awareness of diagnosis) were excluded because they did not meet the significance threshold (p<0.05) for retention during the stepwise selection process or did not improve the model fit as per AIC.

Model comparison and refinement utilized indices including AIC, BIC, the Area Under the receiver operating characteristic Curve (AUC), and goodness-of-fit tests such as the Hosmer-Lemeshow test. Practical considerations, such as data availability, cost, and interpretability, were integrated alongside the evaluation of potential interaction terms. Thorough validation, encompassing both internal and external approaches, was deemed critical for developing a robust, generalizable, and clinically useful model.

This comprehensive framework guided our approach in the schizophrenia study. Specifically, we applied backward elimination (starting with all candidate variables and sequentially removing those with p>0.05), compared competing models based on AIC values, and assessed goodness-of-fit using the Hosmer-Lemeshow test. This process resulted in a final, robust 7-variable model with optimized performance.

We used the Strengthening the Reporting of Observational Studies in Epidemiology to guide the writing (S1 Strobe).

## Results

From a total of 1000 files available for abstraction, 840 medical records were reviewed, 303 records met the inclusion criteria and were included in the analysis. The remaining 538 records were excluded due to incomplete information, such as missing details on age, sex, residence, diagnosis or the outcome, **Fig 1**.

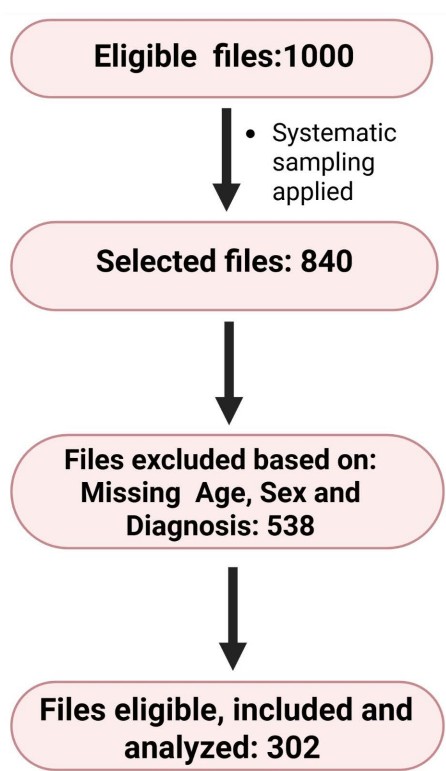

**Fig 1. Eligibility flow diagram.**

## Characteristics of the study population

A total of 303 participants were enrolled in this study, with a median age of 31 years (IQR: 24–41), **Table 1**. The cohort comprised 59.7% males (n = 181) and 40.3% females (n = 122). Bivariate analyses revealed numerous significant associations with illicit substance use. Females exhibited substantially lower illicit substance use prevalence than males (10.7% vs. 44.8%, p < 0.0001). Married participants had significantly lower illicit substance use rates compared to unmarried individuals (17.3% vs. 43.2%, p < 0.0001). Those with both parents deceased showed reduced illicit substance use (18.0% vs. 34.9% in others, p = 0.011), while participants with negative symptoms had lower rates than those without (19.2% vs. 33.9%, p = 0.038). Illicit substance use was strongly associated with alcohol use (60.2% vs. 13.7% in non-users, p < 0.0001), tobacco smoking (61.5% vs. 15.1% in non-smokers, p < 0.0001), and violent behavior (67.0% vs. 38.5% in non-violent participants, p < 0.0001). Medication non-adherence was linked to higher illicit substance use (46.0% vs. 16.4% in adherent individuals, p < 0.0001), while awareness of one's condition was associated with lower illicit substance use (21.2% vs. 39.5% in unaware individuals, p = 0.0006). The illciti substance use group had significantly more hospitalizations (median = 3, IQR:2–4) than the non-use group (median = 2, IQR:1–3, p = 0.044), and younger age was associated with illcit substance use (p = 0.023). Participants with suicidal ideation (n = 3) all had illicit substance use (p = 0.029).

## Assessment of collinearity between variables

Firstly, the correlation matrix performed showed no high correlations (>0.7) between variables, which is good for avoiding multicollinearity, **Fig 2**.

For a more precise assessment of collinearity, we calculated the Variance Inflation Factor (VIF) (**S1 Table**) for fitting a logistic regression model and performed the Hosmer-Lemeshow goodness-of-fit test. The Hosmer-Lemeshow test indicated a good model fit (p = 0.3794 > 0.05), meaning we fail to reject the null hypothesis of adequate fit, **S2 Table**. The model showed good discriminative ability with 83% accuracy, pseudo R² of 0.46, and reasonable precision/recall for both classes.

## Factors associated with substance use in logistic regression

We performed a model comparison and cross-validation comparison for the best logistic regression model for the variables in the study. The backward elimination model performed best overall with the lowest AIC, though the Hosmer-Lemeshow test (Pseudo R-squared of 0.3863; AIC of 246.33 (lowest among all models) and Cross-validation accuracy of 75.93%) suggested poor calibration (p = 0.0196) that might be addressed with additional model refinements, **S3 Table**.

The analysis revealed several factors significantly associated with illicit substance use, both in the initial unadjusted models (OR) and after adjusting for other variables (AOR), **Table 2**. Sex demonstrated a strong protective effect. In the unadjusted model, the odds of substance use were significantly lower for females compared to males (OR = 0.14, 95% CI: 0.07-0.28, p < 0.0001). This association remained significant, though attenuated, in the adjusted model (AOR = 0.27, 95% CI: 0.12-0.60, p = 0.001). Similarly, being Married was associated with substantially lower odds of illcit substance use initially (OR = 0.27, 95% CI: 0.16-0.47, p < 0.0001), and this association persisted even after adjustment (AOR = 0.39, 95% CI: 0.19-0.79, p = 0.008).

The status of having Both Parents Deceased also emerged as associated with lower odds of illicit substance use. The unadjusted model showed lower odds (OR = 0.41, 95% CI: 0.20-0.83, p = 0.013), and interestingly, this association became stronger in the adjusted model (AOR = 0.29, 95% CI: 0.11-0.75, p = 0.011). Higher levels of being Educated were associated with moderately lower odds of illicit substance use in the unadjusted analysis (OR = 0.62, 95% CI: 0.40-0.95, p = 0.029). After adjustment, this association increased in magnitude and significance (AOR = 0.44, 95% CI: 0.24-0.81, p = 0.009).

Conversely, Alcohol Use was strongly associated with illicit substance use. Individuals using alcohol had dramatically higher odds of illicit substance use in the unadjusted model (OR = 9.53, 95% CI: 5.44-16.67, p < 0.0001). Although this

**Table 1. Characteristics of the study populations.**

| Variable | Median (IQR) or Frequency (%) | No Illicit Substance Use % (n) [95% CI] | Yes Illicit Substance Use % (n) [95% CI] | P. value |
|---|---|---|---|---|
| *Age (years)* | 31 (24,41) | 68.9 (208) | 31.1 (94) | **0.023** |
| *Sex, n=301* | | | | |
| Male | *59.7 (181)* | 55.3 (100) [47.7, 62.6] | 44.8 (81) [37.4, 52.3] | **<0.0001** |
| Female | *40.3 (122)* | 89.3 (109) [82.7, 94.2] | 10.7 (13) [5.8, 17.3] | |
| *Marital Status, n=301* | | | | |
| Married | *46.2 (139)* | 82.7 (115) [75.6, 88.5] | 17.3 (24) [11.5, 24.4] | |
| Not Married | *53.8 (162)* | 56.8 (92) [48.9, 64.4] | 43.2 (70) [35.6, 51.1] | **<0.0001** |
| *Educated, n=300* | | | | |
| Primary | *47.7 (143)* | 63.6 (91) [55.3, 71.4] | 36.4 (52) [28.6, 44.7] | 0.074 |
| Secondary | *46.7 (140)* | 71.4 (100) [63.3, 78.7] | 28.6 (40) [21.3, 36.7] | |
| Tertiary | *5.7 (17)* | 88.2 (15) [63.6, 98.5] | 11.8 (2) [1.5, 36.4] | |
| *Employment Status, n=292* | | | | |
| Unemployed | *69.6 (204)* | 66.2 (135) [59.3, 72.6] | 33.8 (69) [27.4, 40.7] | 0.080 |
| Employed | *30.4 (88)* | 76.4 (68) [66.9, 84.8] | 23.6 (21) [15.2, 33.1] | |
| *Residence, n=297* | | | | |
| Urban | *82.8 (246)* | 68.7 (169) [62.6, 74.3] | 31.3 (77) [25.7, 37.4] | 0.587 |
| Rural | *17.17 (51)* | 72.6 (37) [58.3, 84.1] | 27.4 (14) [15.9, 41.7] | |
| *Both Parents Alive, n=296* | | | | |
| No | *58.8 (174)* | 70.7 (123) [63.4, 77.3] | 29.3 (51) [22.7, 36.6] | 0.431 |
| Yes | *41.2 (122)* | 66.4 (81) [57.3, 74.7] | 33.6 (41) [25.3, 42.7] | |
| *Both Parents Deceased, n=296* | | | | |
| No | *79.4 (235)* | 65.1 (153) [58.7, 71.1] | 34.9 (82) [28.9, 41.3] | **0.011** |
| Yes | *20.6 (61)* | 82.0 (50) [70.0, 90.6] | 18.0 (11) [9.4, 30.0] | |
| *Hallucinations, n=301* | | | | |
| No | *18.3 (55)* | 69.1 (38) [55.2, 80.9] | 30.9 (17) [19.1, 44.8] | 0.954 |
| Yes | *81.7 (246)* | 68.7 (169) [62.6, 74.3] | 31.3 (77) [25.7, 37.4] | |
| *Delusions,* | | | | |
| No | *41.9 (127)* | 69.3 (88) [60.6, 77.1] | 30.7 (39) [22.9, 39.4] | 0.919 |
| Yes | *58.1 (176)* | 68.7 (121) [61.4, 75.4] | 31.3 (55) [24.6, 38.6] | |
| Disorganized Speech, *n=300* | | | | |
| No | 43.7 (131) | 74.0 (97) [65.7, 81.2] | 26.0 (34) [18.8, 34.3] | 0.096 |
| Yes | 56.3 (169) | 65.1 (110) [57.4, 72.2] | 34.9 (59) [27.8, 42.6] | |
| Disorganized Behaviour, *n=302* | | | | |
| No | 8.3 (25) | 72.0 (18) [50.6, 87.9] | 28.0 (7) [12.1, 49.4] | 0.724 |
| Yes | 91.7 (277) | 68.6 (190) [62.8, 74.0] | 31.4 (87) [26.0, 37.2] | |
| Negative Symptoms, *n=300* | | | | |
| No | 82.7 (248) | 66.1 (164) [59.9, 71.9] | 33.9 (84) [28.1, 40.1] | *0.038* |
| Yes | 17.3 (52) | 80.8 (42) [67.5, 90.4] | 19.2 (10) [9.6, 32.5] | |
| Cognitive Impairment, *n=300* | | | | |
| No | 39 (117) | 75.2 (88) [66.4, 82.6] | 24.8 (29) [17.4, 33.6] | 0.062 |
| Yes | 61 (183) | 65.0 (119) [57.6, 71.9] | 35.0 (64) [28.1, 42.4] | |
| Suicidal, | | | | |
| No | 99.0 (300) | 69.7 (209) [64.1, 74.9] | 30.3 (91) [25.1, 35.9] | *0.029* |
| Yes | 0.1 (3) | 0.0 (0) [0.0, 70.8] | 100.0 (3) [29.2, 100.0] | |

*(Continued)*

**Table 1.** (Continued)

| Variable | Median (IQR) or Frequency (%) | No Illicit Substance Use % (n) [95% CI] | Yes Illicit Substance Use % (n) [95% CI] | P. value |
|---|---|---|---|---|
| History of Self Harm, *n = 302* | | | | |
| *No* | 93 (281) | 68.7 (193) [63.0, 74.0] | 31.3 (88) [26.0, 37.0] | *0.793* |
| *Yes* | 7 (21) | 71.4 (15) [47.8, 88.7] | 28.6 (6) [11.3, 52.2] | |
| Family History of Schizophrenia | | | | |
| *No* | 97 (294) | 68.7 (202) [63.1, 73.9] | 31.3 (92) [26.1, 36.9] | *0.725* |
| *Yes* | 3 (9) | 77.8 (7) [40.0, 97.2] | 22.2 (2) [2.8, 60.0] | |
| Family History of Other Psychiatric Illness, *n = 294* | | | | |
| *No* | 75.3 (225) | 71.6 (161) [65.2, 77.4] | 28.4 (64) [22.6, 34.8] | *0.128* |
| *Yes* | 24.7 (74) | 62.1 (46) [50.1, 73.0] | 37.9 (28) [27.0, 49.9] | |
| Alcohol Use | | | | |
| *No* | 62.7 (190) | 86.3 (164) [80.5, 90.9] | 13.7 (26) [9.1, 19.5] | ***<0.0001*** |
| *Yes* | 37.3 (113) | 39.8 (45) [30.7, 49.5] | 60.2 (68) [50.5, 69.3] | |
| Violent Behaviour, *n = 302* | | | | |
| *No* | 52.6 (159) | 61.5 (128) [53.8, 68.9] | 38.5 (80) [31.1, 46.2] | **<0.0001** |
| *Yes* | 47.4 (143) | 33.0 (31) [25.4, 41.3] | 67.0 (96) [58.7, 74.6] | |
| Tobacco Smoking, *n = 302* | | | | |
| *No* | 65.6 (198) | 84.9 (168) [79.1, 89.5] | 15.1 (30) [10.5, 20.9] | **<0.0001** |
| *Yes* | 34.4 (104) | 38.5 (40) [29.0, 48.7] | 61.5 (64) [51.3, 71.0] | |
| Aware Of Condition, *n = 299* | | | | |
| *No* | 54.2 (162) | 60.5 (98) [52.5, 68.1] | 39.5 (64) [31.9, 47.5] | ***0.0006*** |
| *Yes* | 45.8 (137) | 78.8 (108) [71.1, 85.3] | 21.2 (29) [14.7, 28.9] | |
| Consulted Traditional Healers, *n = 241* | | | | |
| No | 83.0 (200) | 68.0 (136) [61.1, 74.4] | 32.0 (64) [25.6, 38.9] | 0.514 |
| Yes | 17.0 (41) | 73.2 (30) [57.1, 85.8] | 26.8 (11) [14.2, 42.9] | |
| Patient Adherent to Medication, *n = 293* | | | | |
| *No* | 48.1 (141) | 54.0 (76) [45.4, 62.4] | 46.0 (65) [37.6, 54.6] | ***<0.0001*** |
| *Yes* | 51.9 (152) | 83.6 (127) [76.7, 89.1] | 16.4 (25) [10.9, 23.3] | |
| Number of hospitalizations | 2 (2, 3) | 2 (1, 3) | 3 (2, 4) | **0.044** |

association lessened somewhat in the adjusted model, it remained highly significant and strong (AOR = 6.08, 95% CI: 3.14-11.78, p < 0.0001). Tobacco smoking exhibited a similarly strong association. The unadjusted odds were very high (OR = 8.96, 95% CI: 5.15-15.59, p < 0.0001), and while reduced, the association remained large and highly significant after adjustment (AOR = 4.80, 95% CI: 2.44-9.46, p < 0.0001).

The Number of hospitalizations showed a trend towards being associated with illicit substance use in the unadjusted model but did not reach conventional statistical significance (OR = 1.16, 95% CI: 0.98-1.38, p = 0.076). However, after adjusting for other factors, a higher number of hospitalizations became a statistically significant variable associated with increased odds of illicit substance use (AOR = 1.29, 95% CI: 1.01-1.65, p = 0.038).

Several factors were significant in the unadjusted model but were not retained or reported in the final adjusted model presented. Exhibiting Violent Behaviour was associated with substantially higher odds of illicit substance use (OR = 3.25, 95% CI: 1.94-5.43, p < 0.0001). Higher levels of Negative Symptoms appeared to have an association with lower odds of illicit substance use (OR = 0.46, 95% CI: 0.22-0.97, p = 0.041). Being Aware of Condition was associated with significantly lower odds of illicit substance use (OR = 0.41, 95% CI: 0.25-0.69, p = 0.0008). Adherence to medication was associated

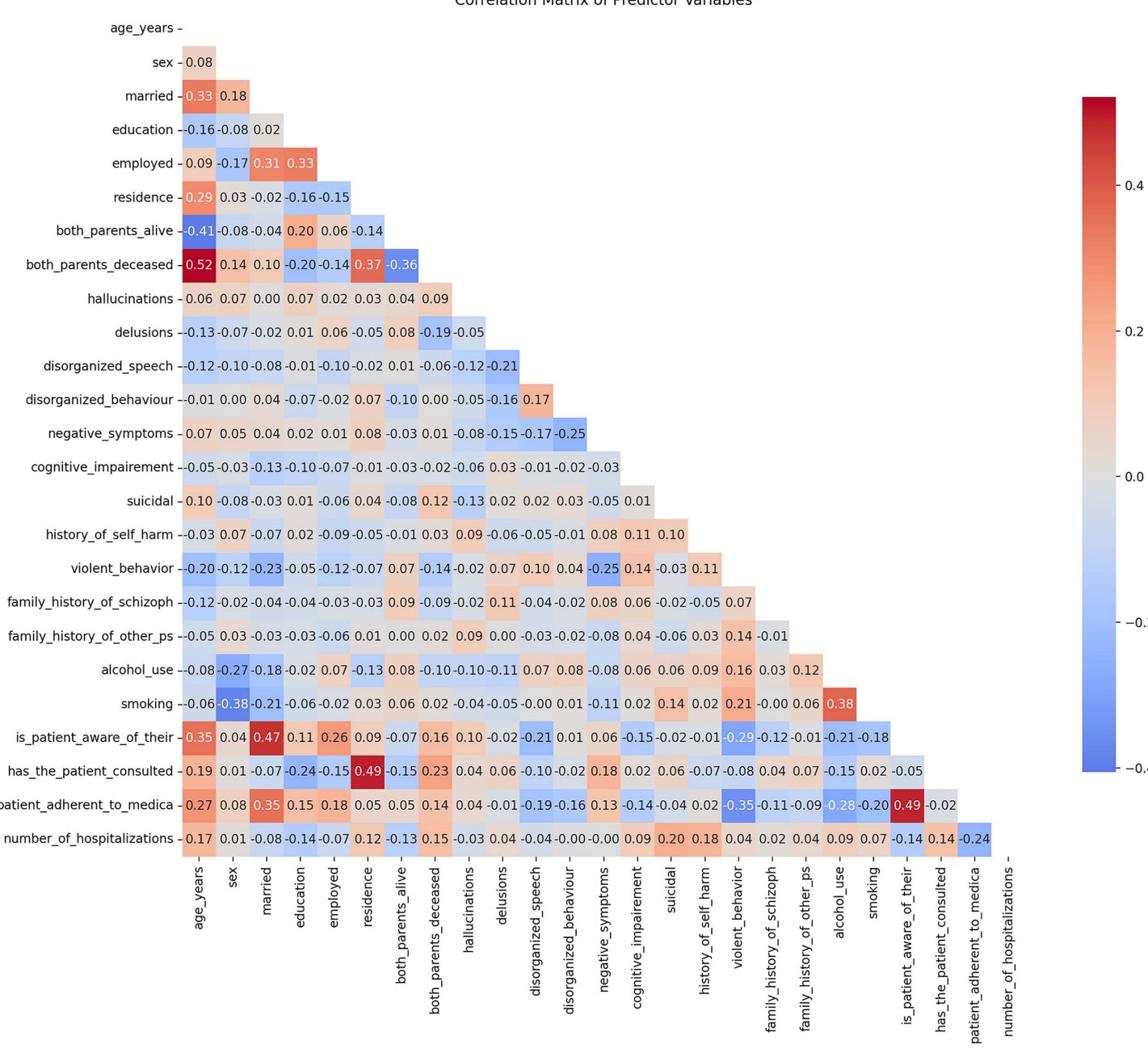

**Fig 2. Correlation matrix of variables.**

with lower odds of illicit substance use in the unadjusted analysis (OR = 0.23, 95% CI: 0.134-0.39, p < 0.0001). Finally, increasing Age was associated with slightly lower odds of illicit substance use in the unadjusted model (OR = 0.97, 95% CI: 0.95-0.99, p = 0.020).

In summary, the variables most strongly associated with illicit substance use in the adjusted model were alcohol use and tobacco smoking. Variables associated with lower odds of illicit substance use in the adjusted model included being

**Table 2. Logistic regression analysis for Substance use.**

| Variables | OR (95%CI) | p. value | AOR (95%CI) | p. value |
|---|---|---|---|---|
| Sex | 0.14 (0.07,0.28) | **<0.0001** | 0.27 (0.12, 0.60) | **0.001** |
| Married | 0.27 (0.16,0.47) | **<0.0001** | 0.39 (0.19, 0.79) | **0.008** |
| Both Parents Deceased | 0.41 (0.20,0.83) | **0.013** | 0.29 (0.11,0.75) | **0.011** |
| Educated | 0.62 (0.40, 0.95) | **0.029** | 0.44 (0.24, 0.81) | **0.009** |
| Alcohol Use | 9.53 (5.44,16.67) | **<0.0001** | 6.08 (3.14,11.78) | **<0.0001** |
| Smoking | 8.96 (5.15,15.59) | **<0.0001** | 4.80 (2.44,9.46) | **<0.0001** |
| Number of hospitalizations | 1.16 (0.98,1.38) | 0.076 | 1.29 (1.01,1.65) | **0.038** |
| Violent Behaviour | 3.25 (1.94, 5.43) | **<0.0001** | | |
| Negative Symptoms | 0.46 (0.22,0.97) | **0.041** | | |
| Aware of Condition | 0.41 (0.25,0.69) | **0.0008** | | |
| Adherence to medication | 0.23 (0.134,0.39) | **<0.0001** | | |
| Age | 0.97 (0.95,0.99) | **0.020** | | |

female, being married, having both parents deceased, and higher education. An increased number of hospitalizations was also associated with illicit substance use after adjustment. Factors like violent behaviour, negative symptoms, awareness of condition, medication adherence, and age were significant in the bivariate analysis but were not included in the final adjusted model (did not retain significance and did not meet the criteria for inclusion in the final model).

## Discussion

The identification of alcohol use, tobacco smoking, and male sex as factors independently associated with illicit substance use in schizophrenia underscores critical intersections between behavioural, biological, and social factors in this population. These findings align with existing literature that positions illicit substance use as both a coping mechanism and a contributor to the complex psychopathology of schizophrenia. The substance use prevalence of 31.1% in our cohort is lower than SSA averages (58–65%) [13] but aligns with Nigerian (43%) [14] and Ethiopian (46.8%) [15] studies. This discrepancy may reflect Zambia's strict substance criminalization or under detection in records. Crucially, alcohol and tobacco smoking were dominant correlates, mirroring global patterns where 60–80% of persons with schizophrenia use tobacco [16] and 30–50% misuse alcohol [17]. Neurobiological mechanisms such as nicotine's transient amelioration of cognitive deficits [18,19] and socioeconomic drivers like *kachasu* affordability likely underpin this. Alcohol and tobacco may serve dual roles: temporarily alleviating symptoms such as anxiety or cognitive deficits while perpetuating cycles of addiction through neurobiological pathways [20,21]. The prominence of these substances highlights the need to address polysubstance use as a multifaceted challenge rather than isolated behaviors.

### Factors associated with lower odds of illicit substance use in multivariable analysis

The identification of female sex, marriage, higher education, and parental loss as factors associated with lower odds of illicit substance use in adjusted models reveals nuanced socio-familial buffers against substance use. The robust association of female sex (AOR = 0.27) aligns with global epidemiological patterns [22,23] but contrasts with Jordanian data where male gender dominated risk profiles [24]. This underscores gender-specific social norms in Zambia, where cultural stigmatization may disproportionately deter illicit substance use among women. Marriage retained significance after adjustment (AOR = 0.39), suggesting spousal support may mitigate isolation-driven illicit substance use, though this effect is attenuated when illicit substance-related behaviours are entrenched. Unexpectedly, having both parents deceased was associated with lower odds of illicit substance use (AOR = 0.29), potentially reflecting early resilience-building or alternative kinship support, a finding warranting qualitative exploration [10,25].

 

The persistence of male sex as a factor associated with illicit substance use raises important questions about gender-specific vulnerabilities. A wide confidence interval for sex variable reflects limited female illicit substance users (n = 13), reducing precision. Despite this, the strong effect size (AOR > 5) and statistical significance (p = 0.003) support male gender as a key correlate. Biological differences, such as hormonal influences on dopamine regulation, may be linked to higher addiction rates. Sociocultural factors, including gendered norms around substance use and disparities in healthcare-seeking behaviour, likely further amplify this association [6,25]. This duality suggests that interventions must extend beyond clinical settings to address broader societal influences, particularly in communities where male illicit substance use is normalized.

### Novel correlate: hospitalization frequency

The emergence of hospitalization frequency as a variable associated with illicit substance use (AOR = 1.29 per admission) signals a clinically critical pattern. This aligns with several studies linking repeated hospitalizations to severe dual-diagnosis trajectories [23,26], suggesting that each admission may represent accumulated neurobiological vulnerability or systemic failures in continuity of care. In LUTH's resource-constrained setting, this association may also reflect cyclical crises where illicit substance use precipitates relapse, driving rehospitalization, a feedback loop demanding integrated inpatient substance interventions.

### Dissociation of bivariate predictors

The loss of significance for violent behavior, negative symptoms, awareness of illness, and medication adherence in adjusted models highlights confounding pathways. For instance, violence (OR=3.25, p < 0.0001 bivariately) may stem from substance-induced impulsivity rather than schizophrenia psychopathology, consistent with neurobiological models of addiction-psychosis interactions [27,28]. Similarly, medication non-adherence likely operates downstream of substance use (e.g., cognitive impairment from alcohol), explaining its attenuation when substance behaviours are modelled. The non-retention of negative symptoms contradicts the "self-medication" hypothesis [28], implying transient symptom alleviation by substances does not translate to population-level protection.

### Clinical and policy implications

The association of education (AOR = 0.44) advocates for socioeconomic interventions like vocational training to disrupt poverty-substance use cycles. Similarly, marital stability and gender-specific associations necessitate family-inclusive counselling and male-targeted harm reduction. The hospitalization finding demands systemic reforms: embedding addiction specialists in inpatient psychiatry teams and post-discharge monitoring. It is important to note that the final logistic regression model showed evidence of poor calibration (Hosmer-Lemeshow p = 0.0196), which may affect the reliability of the predicted probabilities. However, the model demonstrated good discriminative ability. Limitations include recall bias in hospitalization documentation and ASSIST's potential cultural insensitivity to local substances like *kachasu*. Furthermore, continuous predictors like age and number of hospitalizations were modeled as linear terms; their relationship with illicit substance use may be non-linear, an assumption that was not tested and represents a potential limitation. Finally, the cross-sectional design precludes causal inference.

## Conclusion

This study found a prevalence of illicit substance use of 31.1% among persons with schizophrenia at LUTH. Male gender, alcohol use, and tobacco smoking were associated with illicit substance use, but multivariable analysis reveals associations for female sex, marriage, parental loss, and education with lower odds of use, alongside hospitalization frequency as a novel correlate. To mitigate this comorbidity, we recommend: (1) integrating gender-specific substance screening in psychiatric care, (2) socioeconomic programs targeting education and employment, (3) family-centered interventions

 

leveraging marital and kinship support, and (4) specialized addiction services for frequently hospitalized patients. Validating assessment tools for Zambian cultural contexts and longitudinal studies to clarify causality are essential next steps.

## Supporting information

**S1 Checklist. Strobe checklist.**
(DOCX)

**S1 Table. Variance inflation factor for the variables.**
(DOCX)

**S2 Table. Logistic regression model and statistical performance metrics.**
(DOCX)

**S3 Table. Detailed best model performance model, cross validation and calibration metrics.**
(DOCX)

**S1 Data. Minimal dataset.**
(XLSX)

## Author contributions

**Conceptualization:** Kimberley R. Kurehwatira, Sepiso K. Masenga.

**Data curation:** Sepiso K. Masenga.

**Formal analysis:** Sepiso K. Masenga.

**Investigation:** Kimberley R. Kurehwatira, Sepiso K. Masenga.

**Methodology:** Kimberley R. Kurehwatira, Sepiso K. Masenga.

**Project administration:** Sepiso K. Masenga.

**Resources:** Sepiso K. Masenga.

**Supervision:** Joreen P. Povia, Sepiso K. Masenga.

**Validation:** Kimberley R. Kurehwatira, Emmanuel L. Luwaya, Joreen P. Povia, Chileleko Siakabanze, Salma M. Baines, Emmanuel Yumba, Prince Mulambo, David N. Masta, Nestorine N. Ngongo, Natasha Chishala, Emmanuel O. Riwo, Katongo Mutengo, Hanzooma Hatwiko, Martin Chakulya, Lukundo Siame, Bislom C. Mweene, Sepiso K. Masenga.

**Visualization:** Kimberley R. Kurehwatira, Emmanuel L. Luwaya, Joreen P. Povia, Chileleko Siakabanze, Salma M. Baines, Emmanuel Yumba, Prince Mulambo, David N. Masta, Nestorine N. Ngongo, Natasha Chishala, Emmanuel O. Riwo, Katongo Mutengo, Hanzooma Hatwiko, Martin Chakulya, Lukundo Siame, Bislom C. Mweene, Sepiso K. Masenga.

**Writing – original draft:** Kimberley R. Kurehwatira, Emmanuel L. Luwaya, Joreen P. Povia, Chileleko Siakabanze, Salma M. Baines, Emmanuel Yumba, Prince Mulambo, David N. Masta, Nestorine N. Ngongo, Natasha Chishala, Emmanuel O. Riwo, Katongo Mutengo, Hanzooma Hatwiko, Martin Chakulya, Lukundo Siame, Bislom C. Mweene, Sepiso K. Masenga.

**Writing – review & editing:** Kimberley R. Kurehwatira, Emmanuel L. Luwaya, Joreen P. Povia, Chileleko Siakabanze, Salma M. Baines, Emmanuel Yumba, Prince Mulambo, David N. Masta, Nestorine N. Ngongo, Natasha Chishala, Emmanuel O. Riwo, Katongo Mutengo, Hanzooma Hatwiko, Martin Chakulya, Lukundo Siame, Bislom C. Mweene, Sepiso K. Masenga.

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
