## [Decision Letter · Decision Letter 0]

6 Jun 2025

PGPH-D-25-01149

Prevalence and Factors Associated with Substance Abuse Among Schizophrenic Patients at LUTH in Zambia

Dear Dr. Masenga,

Thank you for submitting your manuscript to PLOS Global Public Health. After careful consideration, we feel that it has merit but does not fully meet PLOS Global Public Health’s publication criteria as it currently stands. Therefore, we invite you to submit a revised version of the manuscript that addresses the points raised during the review process.

We look forward to receiving your revised manuscript.

Kind regards,

Feten Fekih-Romdhane

Academic Editor

Journal Requirements:

1. Please insert an Ethics Statement at the beginning of your Methods section, under a subheading 'Ethics Statement'.

2. We note that your Data Availability Statement is currently as follows: “The raw data underlying the results presented in the study have been uploaded as supporting information.”.

Additional Editor Comments (if provided):

Reviewers' comments:

Reviewer's Responses to Questions

**Comments to the Author**

1. Does this manuscript meet PLOS Global Public Health’s publication criteria?

Reviewer #1: Yes

Reviewer #2: Yes

Reviewer #3: Yes

2. Has the statistical analysis been performed appropriately and rigorously?

Reviewer #1: Yes

Reviewer #2: No

Reviewer #3: Yes

3. Have the authors made all data underlying the findings in their manuscript fully available (please refer to the Data Availability Statement at the start of the manuscript PDF file)?

Reviewer #1: Yes

Reviewer #2: Yes

Reviewer #3: Yes

4. Is the manuscript presented in an intelligible fashion and written in standard English?

Reviewer #1: Yes

Reviewer #2: Yes

Reviewer #3: Yes

Reviewer #1: Overall: I would like to thank for inviting to review this manuscript. This study will have great value in sub-Saharan countries to enhance awareness and prevention strategies for mental health wellbeing and its association with substance use disorders. I hope this manuscript described well.

I have few comments, find it below

Comments

Abstract section

1. The word alcohol (kachasu) mentioned, what does ‘kachasu’ mean? Is it Zambian language for alcohol or not? If you misspelled kaht use revise it

2. DSM-5 or DSM-5TR currently defines substance use rather than substance abuse. Since the word abuse is inappropriate term and increase negative view on substance users. It is better to replace substance abuse by substance use in your whole document

3. ‘Ethical approval and informed consent were obtained’ put this statement in ethical consideration section only.

Introduction/Background

1. ‘Yet the country has only five psychiatrists’. This statement is for whole Zambian country or the only for University Teaching Hospital (LUTH). Make clear this statement otherwise it seems like exaggeration of the problems

Method

1. The data collected from patient medical recorded or secondary data might be prone to biased information, what you did to minimize bias in your study? Describe this carefully in the limitation section

2. Describe smartly about the data quality assurance since the secondary data need more attention to minimize missed information.

3. What you did for participants who had substance use disorder after screening? If you did something, please specify it?

Result section

1. For the variables Age and Number of hospitalizations has reference category not specified why? Use the same format for all variables.

Discussion

1. ‘In this study the prevalence of substance abuse was 31.1% (94/303). Alcohol use (60.2% vs.46 13.7% non-users, p<0.001) and smoking (61.5% vs. 15.1% non-smokers, p<0.001)’. Please discuss this finding with others study which have similar, higher or below finding compared to this study by giving convincing scientific evidences

Limitations, Recommendation and conclusion

1. Describe limitation in detail

2. Conclusion Should be smart, short and prices

3. Recommendation based on finding, be carefully about this

4. Is it ASSIT valid in Zambia? If not valid it is better to recommend future study to do validation.

Overall: be carefully about grammar, paraphrasing and correct reference

Finally; - thank you for your contribution to existing body of knowledge’s and submitting your manuscript to PLOS Global Public Health

Reviewer #2: Dear Author,

Thanks for submitting the article.

Following are my suggestions :

1. Clearly define "substance abuse" to align with ICD-10, including alcohol (F10) and nicotine (F17). Ensure consistency across methodology and results to avoid confusion.

2. Include a power calculation to justify the sample size of 302 participants. This is necessary to confirm the study’s ability to detect associations. Eg for rare variables like suicidal behavior (n=3).

3. Analyze the 538 excluded records (64% of 840) to assess systematic differences from included cases. High exclusion rates may introduce bias.

4. Provide details on the quality and consistency of medical record documentation. Specify any quality checks or standardization processes applied.

5. State the criteria for selecting variables for multivariable logistic regression, such as p<0.05 in univariable analysis or theoretical relevance.

6. Assess collinearity between predictors, such as alcohol use and smoking, using variance inflation factor (VIF) or correlation matrices. Report goodness-of-fit with tests like Hosmer-Lemeshow.

7. Explain wide confidence intervals, such as for male sex (AOR=5.21, 95% CI: 1.75–15.63).

8. Limit the scope of results to tertiary settings similar to LUTH. From a single-center study, this cannot be claimed as “nationally representative data”.

Thank you

Regards

Reviewer #3: General comments and some key concerns:

The manuscript is addressing an important issue on “Prevalence and Factors Associated with Substance Abuse among Schizophrenic Patients at LUTH in Zambia”. Substance Abuse and use has remained a global challenge in the population including the Schizophrenic Patients with limited data on the extent of their use among this population. However, there are some comments that need to be addressed as below.

• The authors should use past tense when reporting in the manuscript and some sentences are too long and hence losing the intended meaning. The sentence in line 19-21 need to be split up to get a clear meaning.

1. Abstract

The authors should include a sub-section on the aim or objectives of the study just after the background. Authors should also include the summary of data analysis plan in the methods.

2. Introduction

In the line 66, the authors should insert Introduction instead of Background. The authors should incorporate the objective(s) of the study at the end of the sub-section

3. Methods section

The study design is not clear and the authors should think through it since in the abstract it is a different design. The authors should include a sub-section on the study population i.e. target population, accessible population and the actual study population recruited in the study just before eligibility and recruitment. Furthermore, the study populations are schizophrenic patients and they have lost reality of the world and therefore the responses they give may not be well connected. Which type of schizophrenic patients’ records were used or participated in the study? In the abstract, data was collected using structured interviews, validated tools (Alcohol, Smoking, and Substance Involvement Screening Test [ASSIST]), but in the methods medical records were used, how were interviews made on the records?

4. Results

In table 1, demographic characteristics, the authors should include the units of age, and since the main measure is % and not frequencies, the authors should use % (n) and not the reverse. The total number of participants was 302, but in table 1, some items responses exceed that number. The authors should check this throughout the table. In table2, there is a strong association of Violent Behavior and substance use, how do violent behaviors lead to substance abuse?

5. Discussion

The authors should split the discussion if possible based on the study objectives so that it is clear i.e. demographic characteristics, substance use and associated risk factors.

6. Conclusion

The authors should have a sub-section of conclusions in the manuscript and not entangled in the discussion. And the conclusion made neither is nor supported by the findings of the study and therefore it needs to be re-written.

6. References

Further citations are needed to be included to support the background to the study and the authors should also follow the recommended Journal reference style.

**Do you want your identity to be public for this peer review?** For information about this choice, including consent withdrawal, please see our Privacy Policy

Reviewer #1: **Yes: ** Tamene Berhanu Alaho

Reviewer #2: **Yes: ** Karthik Balajee Laksham

Reviewer #3: No

---

## [Decision Letter · Decision Letter 1]

4 Sep 2025

PGPH-D-25-01149R1

Prevalence and Factors Associated with Substance use Among Schizophrenic Patients at a Tertiary Referral Hospital in Zambia

Dear Dr. Masenga,

Thank you for submitting your manuscript to PLOS Global Public Health. After careful consideration, we feel that it has merit but does not fully meet PLOS Global Public Health’s publication criteria as it currently stands. Therefore, we invite you to submit a revised version of the manuscript that addresses the points raised during the review process.

EDITOR:

The manuscript contains causal language (e.g., “predictors”), which is inappropriate given the cross-sectional design. Such statements are misleading and should be removed. The authors should revise these statements to reflect correlational findings.

The use of labeling and stigmatizing language is not accepted. Stigmatizing phrases, such as “schizophrenic patients”, should all be removed throughout the paper. The title should also be revised accordingly.

Clear objectives and hypotheses should be formulated at the end of the introduction. Inclusion criteria should be further detailed (outpatients or inpatients? clinical remission? schizophrenia according to which criteria? who made the diagnosis?).

The methods section is poorly informed. More details are required about how clinical variables were assessed (hallucinations, delusions, disorganized speech, disorganized behavior, negative symptoms, cognitive impairment, suicidality and self-harm).

Please submit your revised manuscript by . If you will need more time than this to complete your revisions, please reply to this message or contact the journal office at globalpubhealth@plos.org. Please include the following items when submitting your revised manuscript:

We look forward to receiving your revised manuscript.

Kind regards,

Feten Fekih-Romdhane

Academic Editor

Journal Requirements:

Additional Editor Comments (if provided):

Reviewer #1:

Reviewer #3:

Reviewers' comments:

Reviewer's Responses to Questions

**Comments to the Author**

Reviewer #1: All comments have been addressed

Reviewer #3: All comments have been addressed

publication criteria?

Reviewer #1: Yes

Reviewer #3: Yes

3. Has the statistical analysis been performed appropriately and rigorously?

Reviewer #1: Yes

Reviewer #3: Yes

4. Have the authors made all data underlying the findings in their manuscript fully available (please refer to the Data Availability Statement at the start of the manuscript PDF file)?

Reviewer #1: Yes

Reviewer #3: Yes

5. Is the manuscript presented in an intelligible fashion and written in standard English?

Reviewer #1: Yes

Reviewer #3: Yes

Reviewer #1: I have no comments

Reviewer #3: General comments and some key concerns:

The authors have tried to address all the comments raised in the paper and congratulate them. However, there still some few issues that needs to be addressed.

1. Abstract

The objective in the abstract should be complete (see the title). The conclusion drawn by the authors in the abstract are not in line with the objective and the title. Therefore, the authors need to paraphrase the conclusion. The authors need to be clear on the sentence (line 35-36), “Data were collected from medical records including records of structured interviews with validated tools…”, were the structured interviews information retrieved also from records or from the medical records, that is when the study participants were contacted for the interview? The authors need to clarify on this.

2. Introduction

The authors need to improve the grammar on the sentence (Line 88-91). And the objective of the study also needs to be improved i.e. stated clearly for easy understanding.

3. Methods section

According to the study design, which is the retrospective cross-sectional study, as mentioned above “Data were collected from medical records including records of structured interviews with validated tools…”, were the structured interviews information retrieved also from records or from the medical records, that is when the study participants were contacted for the interview? The authors need to clarify on this. Line 144, no patient interviews were conducted; the authors need to clear how information from the structured interviews mentioned in the abstract were obtained. Line 121, the authors should avoid starting a sentence with a value.

4. Results

Line 296, the authors mention “smoking”, which smoking is being referred in the paper?

6. Conclusion

The authors should also conclude on the prevalence since it is key in the title.

**Do you want your identity to be public for this peer review?** For information about this choice, including consent withdrawal, please see our Privacy Policy

Reviewer #1: **Yes: ** Tamene Berhanu Alaho

Reviewer #3: No

---

## [Decision Letter · Decision Letter 2]

22 Oct 2025

PGPH-D-25-01149R2

Prevalence and Factors Associated with Substance use Among Persons with Schizophrenia at a Tertiary Referral Hospital in Zambia

Dear Dr. Masenga,

Thank you for submitting your manuscript to PLOS Global Public Health. After careful consideration, we feel that it has merit but does not fully meet PLOS Global Public Health’s publication criteria as it currently stands. Therefore, we invite you to submit a revised version of the manuscript that addresses the points raised during the review process.

We look forward to receiving your revised manuscript.

Kind regards,

Feten Fekih-Romdhane

Academic Editor

Journal Requirements:

Additional Editor Comments (if provided):

Reviewers' comments:

Reviewer's Responses to Questions

**Comments to the Author**

Reviewer #3: All comments have been addressed

Reviewer #4: All comments have been addressed

Reviewer #5: (No Response)

Reviewer #6: All comments have been addressed

Reviewer #7: (No Response)

publication criteria?

Reviewer #3: Yes

Reviewer #4: Partly

Reviewer #5: Yes

Reviewer #6: Yes

Reviewer #7: Yes

3. Has the statistical analysis been performed appropriately and rigorously?

Reviewer #3: Yes

Reviewer #4: Yes

Reviewer #5: I don't know

Reviewer #6: Yes

Reviewer #7: Yes

4. Have the authors made all data underlying the findings in their manuscript fully available (please refer to the Data Availability Statement at the start of the manuscript PDF file)?

Reviewer #3: Yes

Reviewer #4: Yes

Reviewer #5: Yes

Reviewer #6: Yes

Reviewer #7: Yes

5. Is the manuscript presented in an intelligible fashion and written in standard English?

Reviewer #3: Yes

Reviewer #4: No

Reviewer #5: Yes

Reviewer #6: Yes

Reviewer #7: Yes

Reviewer #3: Abstract

Line 29 -31: Objective should be revised as “This study determined the prevalence and factors associated with substance use among adults with schizophrenia at the University Teaching Hospital (LUTH) in Zambia

Introduction

Line 99 -101: The suggested study objective is “The study determined the prevalence and factors associated with substance use among adults with schizophrenia at the University Teaching Hospital (LUTH) in Zambia”.

Results

Line 304: The statement “Conversely, Alcohol Use was strongly associated with substance use”, what does this mean? Alcohol is also a substance!!!

Reviewer #4: Major Revision. The study is important, but clarification of methods, removal of redundancy, and careful editing for consistency are needed.

Reviewer #5: Summary and Overall Assessment

This manuscript is well-crafted and presents meaningful findings. The study is methodologically robust, and the results are clearly articulated. It makes a valuable contribution to the field, enhancing our understanding of the topic. I recommend minor revisions to enhance clarity, formatting, and alignment with the journal's guidelines before final acceptance.

Strengths

• The research objectives are clearly stated, and the methodology is appropriately designed.

• The results are logically organised and supported by relevant data.

• The discussion thoughtfully integrates the findings with existing literature and highlights important public health implications.

Areas for Improvement

- Abstract Length: The abstract currently exceeds the 300-word limit set by PLOS Public Health. Please revise it to be more concise while retaining essential information.

- Reference Formatting: Some references are missing DOI numbers. Kindly verify and update all references to include DOIs where available.

- Language and Style: Minor grammatical and typographical errors are present throughout the manuscript. A thorough proofreading is recommended.

Final Remarks

Overall, this is a well-structured and informative study that offers valuable insights for public health research. With the above minor revisions, the manuscript will be well-positioned for publication in PLOS Public Health.

Reviewer #6: Congratulations to the authors! This is a very well-written paper, and all previous comments have been addressed appropriately.

I have very minor comments for the authors:

1) Clarify that you are assessing psychoactive substance use earlier in the paper, as substance use can sometimes be an umbrella term and include alcohol and/or tobacco use. This can be a bit confusing and the first mention of psychoactive substance use is in the methods.

2) Cite the software used for analysis.

3) Cite the statistical standards and threshold described in the methods.

Other than these, I think this paper is ready.

Reviewer #7: This cross-sectional study explores substance use among persons with schizophrenia in Zambia, addressing a critical evidence gap in both the local and broader sub-Saharan African literature. The study offers valuable insights into a population at high risk for substance use disorders.

Main Strengths

1. The study addresses a clear evidence gap within Zambia’s mental health system.

2. It meets STROBE reporting standards, enhancing transparency and reproducibility.

3. Clinical and sociodemographic variables were appropriately selected and comprehensively assessed.

4. Statistical methods were rigorous, with multivariable modeling, collinearity diagnostics, and model fit procedures clearly reported.

Points for Improvement

1. In broader literature, “substance use” typically includes tobacco, alcohol, and other drugs. Excluding tobacco and alcohol may confuse readers. A more accurate term might be illicit substance use to reflect the study's classification of other drugs.

2. The final logistic regression model demonstrated poor calibration (Hosmer-Lemeshow p = 0.0196), raising concerns about the model’s fit. The reliability of the associations could be improved by re-estimating the model using penalized regression or by reducing the number of predictors.

3. Continuous predictors like age and number of hospitalizations were modeled linearly. However, their relationship with substance use may be non-linear. The analysis would benefit from assessing this assumption using methods such as the Box-Tidwell test or by modeling non-linearity with splines or categorized versions.

4. While adjusted predictors are clearly reported, some variables significant in bivariate analysis (e.g., negative symptoms, awareness of diagnosis) were omitted from the final model without explanation. Clarifying whether these were excluded based on stepwise selection or clinical irrelevance would enhance transparency.

Minor Comments

Descriptive tables should include 95% confidence intervals for proportions to improve interpretability and comparability across groups.

Overall, this is a timely and relevant study. The analytical approach is largely sound and well-reported. Nonetheless, addressing modeling assumptions, clarifying variable inclusion decisions, and improving model calibration would strengthen the robustness of the findings.

**Do you want your identity to be public for this peer review?** For information about this choice, including consent withdrawal, please see our Privacy Policy

Reviewer #3: No

Reviewer #4: **Yes: ** Fathi Ali Araye

Reviewer #5: No

Reviewer #6: No

Reviewer #7: **Yes: ** Honest Anaba

---

## [Decision Letter · Decision Letter 3]

22 Dec 2025

Prevalence and Factors Associated with illicit Substance use Among Persons with Schizophrenia at a Tertiary Referral Hospital in Zambia

PGPH-D-25-01149R3

Dear Prof. Masenga,

We are pleased to inform you that your manuscript 'Prevalence and Factors Associated with illicit Substance use Among Persons with Schizophrenia at a Tertiary Referral Hospital in Zambia' has been provisionally accepted for publication in PLOS Global Public Health.

Best regards,

Yatan Pal Singh Balhara

Academic Editor

Reviewer Comments (if any, and for reference):

Reviewer's Responses to Questions

**Comments to the Author**

Reviewer #4: All comments have been addressed

Reviewer #6: All comments have been addressed

publication criteria?

Reviewer #4: Yes

Reviewer #6: Yes

3. Has the statistical analysis been performed appropriately and rigorously?

Reviewer #4: Yes

Reviewer #6: Yes

4. Have the authors made all data underlying the findings in their manuscript fully available (please refer to the Data Availability Statement at the start of the manuscript PDF file)?

Reviewer #4: Yes

Reviewer #6: Yes

5. Is the manuscript presented in an intelligible fashion and written in standard English?

Reviewer #4: Yes

Reviewer #6: Yes

Reviewer #4: (No Response)

Reviewer #6: I think paper is bridging a major gap in the study of illicit substance use among people with schizophrenia in a resource-constrained environment.

**Do you want your identity to be public for this peer review?** For information about this choice, including consent withdrawal, please see our Privacy Policy

Reviewer #4: **Yes: ** Fathi Araye

Reviewer #6: No
